

# Variable absorption of mutational trends by prion-forming domains during *Saccharomycetes* evolution

Paul M. Harrison

Department of Biology, McGill University, Monteal, Quebec, Canada

## ABSTRACT

Prions are self-propagating alternative states of protein domains. They are linked to both diseases and functional protein roles in eukaryotes. Prion-forming domains in *Saccharomyces cerevisiae* are typically domains with high intrinsic protein disorder (*i.e.,* that remain unfolded in the cell during at least some part of their functioning), that are converted to self-replicating amyloid forms. *S. cerevisiae* is a member of the fungal class *Saccharomycetes*, during the evolution of which a large population of prion-like domains has appeared. It is still unclear what principles might govern the molecular evolution of prion-forming domains, and intrinsically disordered domains generally. Here, it is discovered that in a set of such prion-forming domains some evolve in the fungal class *Saccharomycetes* in such a way as to absorb general mutation biases across millions of years, whereas others do not, indicating a spectrum of selection pressures on composition and sequence. Thus, if the bias-absorbing prion formers are conserving a prion-forming capability, then this capability is not interfered with by the absorption of bias changes over the duration of evolutionary epochs. Evidence is discovered for selective constraint against the occurrence of lysine residues (which likely disrupt prion formation) in *S. cerevisiae* prion-forming domains as they evolve across *Saccharomycetes*. These results provide a case study of the absorption of mutational trends by compositionally biased domains, and suggest methodology for assessing selection pressures on the composition of intrinsically disordered regions.

## INTRODUCTION

Prion formation and propagation has been discovered and investigated chiefly in the budding yeast *Saccharomyces cerevisiae*, which is a member of the fungal class *Saccharomycetes*. The yeast *S. cerevisiae* has >200 prion-like proteins that have N/Q-rich domains of the sort observed in ≥8 known prion-formers (*An, Fitzpatrick & Harrison, 2016*; *Harbi & Harrison, 2014a*; *Harbi et al., 2012*). Such yeast prions have been causally associated with diverse phenomena including evolutionary capacitance, large-scale genetic control, and yeast disease-like conditions. Examples of these proteins in *S. cerevisiae* are reviewed in the introduction to a previous paper (*Su & Harrison, 2019*). In the fission yeast *Schizosaccharomyces pombe* and the fungus *Podospora anserina*, there are also observed

Corresponding author
Paul M. Harrison,
paul.harrison@mcgill.ca

prions (*Saupe, 2011*; *Sideri et al., 2017*). Recently, prion-like proteins have been associated with the formation of membraneless biomolecular condensates, such as stress granules (*Franzmann et al., 2018*; *Jain et al., 2016*).

A bias for the polar residues asparagine (N) and/or glutamine (Q), and high intrinsic disorder are major features of the amyloid-based prion formers (*Harbi & Harrison, 2014b*). Computer programs that discriminate prion-like sequence compositions have been derived for annotating potential prion-forming regions (*Alberti et al., 2009*; *Espinosa Angarica, Ventura & Sancho, 2013*; *Lancaster et al., 2014*; *Ross et al., 2013*; *Zambrano et al., 2015*).

The original mammalian Prion Protein domain does not have N/Q bias, and is conserved deeply since in early chordates a Prion Protein ancestral gene appeared (*Ehsani et al., 2011*; *Harrison, Khachane & Kumar, 2010*; *Westaway et al., 2011*). However, Sup35p (which underlies the [$PSI^+$] prion) has an N/Q bias that is prevalent across the *Ascomycota* and *Basidiomycota* phyla, which had a last common ancestor > 1 billion years ago (*Harrison et al., 2007*). A surge in the emergence of N/Q-rich yeast-prion-like proteins early in *Saccharomycetes* evolution resulted from mutational trends to form more polyasparagine tracts, providing the molecular basis from which several known prion-forming domains seem to have spawned (*An, Fitzpatrick & Harrison, 2016*). Prion-forming domains from *S. cerevisiae* tend to evolve more quickly as sequences than other prion-like domains but maintain their prion-like composition (*Su & Harrison, 2019*). In humans, several yeast-prion-like proteins are implicated in neurodegenerative processes (*Kim et al., 2013*; *Pokrishevsky, Grad & Cashman, 2016*; *Sun et al., 2011*). In *Aplysia* and *Drosophila*, such proteins have been associated with formation and preservation of long-term memory (*Khan et al., 2015*; *Si et al., 2010*). Other eukaryotes, such as *Drosophila melanogaster*, *Plasmodium falciparum* and the leech *Helobdella robusta* are home to substantial sets of prion-like proteins (*An & Harrison, 2016*; *Pallares et al., 2018*). The slime mold *Dictyostelium* has greater than fifth of its proteome displaying prion-like composition (*An & Harrison, 2016*; *Malinovska & Alberti, 2015*), and it maintains a cellular system for avoiding prion-like aggregation and propagation (*Malinovska & Alberti, 2015*; *Malinovska et al., 2015*). In all domains of life, prion-like proteins have been observed (*Espinosa Angarica, Ventura & Sancho, 2013*; *Tetz & Tetz, 2017*; *Tetz & Tetz, 2018*), with many thousands annotated in bacteria (*Harrison, 2019*; *Iglesias, De Groot & Ventura, 2015*). Bacterial prion-forming proteins have been observed experimentally (*Molina-Garcia et al., 2018*; *Shahnawaz et al., 2017*; *Yuan et al., 2014*; *Yuan & Hochschild, 2017*). Hundreds of bacterial prion-like proteins occur across multiple bacterial phyla in a sparse conservation pattern (*Harrison, 2019*).

Here, the evolution of the sequences of prion-forming domains in *Saccharomycetes* is re-visited, but from the point of view of mutation biases. Protein regions are discovered to variably absorb mutation biases that are observable in the proteome as a whole. This is evidenced in the numbers of prion-like proteins, the percentage of guanidine and cytidine (GC%) in the DNA, and the proportions of poly-asparagine and poly-glutamine.

## METHODS

### Data

The reference set of proteomes for *Saccharomycetes* (73 organisms) was sourced in June 2017 from UniProt (*Boeckmann et al., 2003*) (http://www.uniprot.org). Sets of proteins with prion-forming domains (Data S1) and their orthologs across *Saccharomycetes* were collated as previously described (*Su & Harrison, 2019*).

### Prion-like composition

Prion-like composition in orthologs was calculated in two ways, firstly using the PLAAC prion-like domain annotation program (*Lancaster et al., 2014*), and secondly using the fLPS program for annotation of compositional biases (*Harrison, 2017*). These were both run using default parameters, except that for fLPS the expected frequency for glutamine and asparagine residues was set equal to 0.05. For PLAAC, both the PRD score and the LLR score were analysed; the former is an indicator of the overall amount of prion-like composition in an annotated bounded prion-like region, while the latter indicates the prion-like sequence composition of the best sequence window (*Lancaster et al., 2014*). PLAAC scores < 0.0 or labelled 'N/A' in the output are set equal to 0.0 here.

### Measures of proteome bias

Several measures of compositional bias across proteomes/genomes were examined:

(i)   %N (asparagine) in the proteome;
(ii)  %Q (glutamine) in the proteome;
(iii) % poly-N in the proteome (with a minimum tract length of 3);
(iv)  % poly-Q in the proteome (with a minimum tract length of 3);
(v)   % poly-Q + poly-N in the proteome (with a minimum tract length of 3);
(vi)  %GC in the DNA;
(vii) The fraction of N/Q-rich proteins in the proteome according to a specific fLPS bias *P*-value threshold (either 1e−08, 1e−10 or 1e−12);
(viii) The fraction of proteins in the proteome with prion-like composition according to the program PLAAC (with PRD score >0.0, ≥15.0 or ≥30.0, or similarly for LLR score).

Measures (i) to (v) were chosen since there are indicative of general mutational trends that are relevant to the predominant compositional biases of prion-forming domains in *S. cerevisiae*, namely bias towards asparagine and glutamine and tracts of these residue types (*An, Fitzpatrick & Harrison, 2016*). Measure (vi) (%GC) is the most basic compositional trend that can be analyzed for genomic DNA, which might underlie trends at the amino-acid level. Measures (vii) to (viii) indicate the degree to which individual proteins throughout the proteome have prion-like compositional biases to a certain level, and so would indicate how every protein is on average affected by mutational trends.

### Correlations

Both weighted and unweighted Pearson correlation coefficients were calculated to assess the correlations of individual prion-like composition with the general trends in the proteome. Weightings for plot points were calculated according to their closest similarity with another

 

protein, calculated as (1-%**I**/100), where %**I** is the percentage sequence identity in the most significant BLASTP sequence alignment (*Altschul et al., 1997*). These weightings were summed appropriately, as described in previous analyses (*Harrison, 2019*; *Su & Harrison, 2019*). Results indicate that the overall outcomes for specific proteins are not affected by non-usage of such weightings (see below).

## RESULTS

### Initial example: the Ure2 prion-forming domain demonstrates strong absorption of mutational trends

As an initial example, the evolutionary behaviour of compositional biases in the prion-forming domain of Ure2p, which underlies the [*URE3*] prion, was examined (Figs. 1–2). The current data indicate that an ancestor of the Ure2p prion-forming domain with a strong N/Q-rich prion-like composition originated early in *Saccharomycetes* evolution (at least in the last common ancestor of the diverse families *Debaryomycetaceae* and *Saccharomycetaceae*), in agreement with results in previous publications (*An, Fitzpatrick & Harrison, 2016*; *Harrison et al., 2007*) (Fig. 3; the organismal branching pattern from recent fungal phylogenies was used (*Kurtzman & Robnett, 2013*; *Shen et al., 2016*). In general, there is a strong correlation between the degree of bias in the N/Q-rich region of Ure2p and the degree of compositional bias in the whole proteome/genome by several indicators (%polyasparagine or %[polyasparagine + polyglutamine] or DNA GC% or fraction of N/Q-rich prion-like proteins with fLPS $P$-value $< 10^{-10}$) (Fig. 1). The correlations with PLAAC prion-like composition score are lower, but both measures have strong correlations with %GC in DNA (Fig. 2). Thus, during the surge in formation of prion-like regions during *Saccharomycetes* evolution (*An, Fitzpatrick & Harrison, 2016*), the degree of N-bias in the individual prion-former Ure2p also increased in correlation with the general trend as it panned out across various sub-clades.

### Other prion-forming proteins show a variable spectrum of absorption of mutational trends across *Saccharomycetes*

Of the known amyloid-based prions—as well as Ure2p—Swi1p, Cyc8p and Sup35p have domains of prion-like composition or N/Q bias that are widespread across *Saccharomycetes* (in 84% of orthologs for Cyc8p, 98% for Swi1p, and 90% for Sup35p; Table S1), with such domains of these latter three also arising in other *Ascomycota* clades (*An, Fitzpatrick & Harrison, 2016*; *Harrison et al., 2007*).

In general, there are strong correlations for Ure2p, Swip and Cyc8p with %N, %poly-N, %GC in DNA and with the numbers of proteins with prion-like composition (Tables 1–2). Within these general trends, these four demonstrate a spectrum of responses to the overall proteome-wide mutational trends, with Ure2p being the strongest correlator. Sup35p stands out as an exception; it shows on the whole weaker correlations generally with %N and %poly-N, and stronger correlations with %poly-Q than the other three. This may be because there is selection pressure to maintain a specific proportion of Qs in specific local patterns or ratios (*MacLea et al., 2015*).

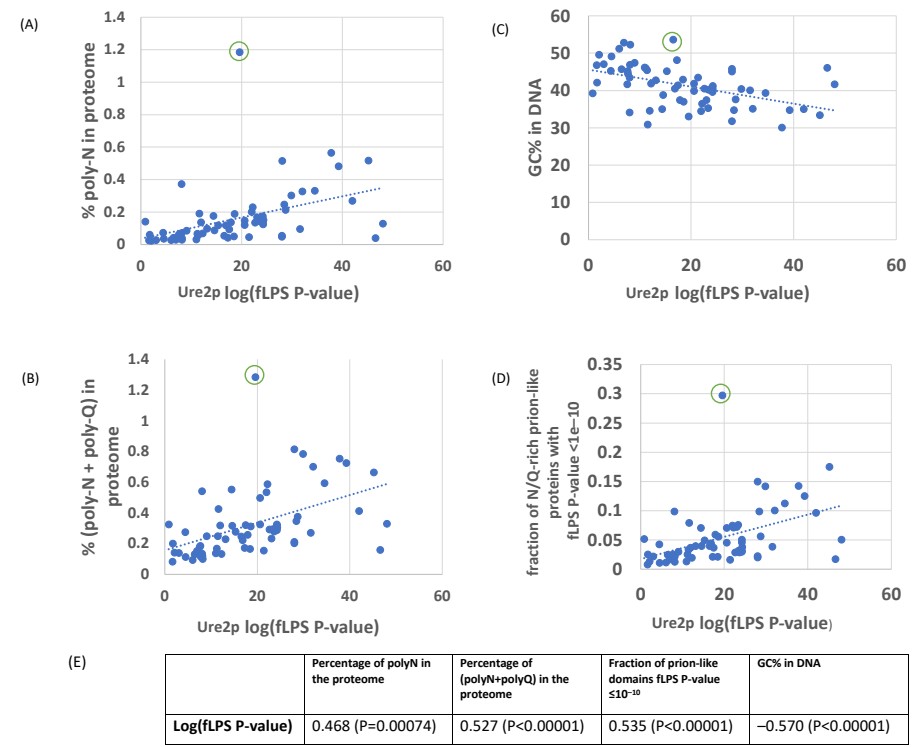

| | Percentage of polyN in the proteome | Percentage of (polyN+polyQ) in the proteome | Fraction of prion-like domains fLPS P-value ≤10⁻¹⁰ | GC% in DNA |
|---|---|---|---|---|
| Log(fLPS P-value) | 0.468 (P=0.00074) | 0.527 (P<0.00001) | 0.535 (P<0.00001) | −0.570 (P<0.00001) |

**Figure 1  Correlation of various measures of mutational bias across proteomes versus the individual compositional bias in the Ure2p prion-forming domain, as judged by the fLPS program.** The outlier proteome *A. rubescens* is ringed. (A) Percentage of poly-N residues in the proteome. (B) Percentage of (poly-N + poly-Q) residues in the proteome. (C) DNA GC%. (D) Fraction of N/Q-rich prion-like proteins with fLPS *P*-value <1e−10. (E) Table of correlations and significances for plots (A) to (D).

Furthermore, Pin3 protein also has a widespread prion-like domain across *Saccharomycetes*, there being 52/55 (95%) *Saccharomycetes* Pin3 orthologs having PLAAC LLR scores > 15.0. However, the degree of conservation of N/Q-rich bias *per se* is lower for this protein with 38/55 (75%) having a fLPS compositional bias *P*-value ≤1e−10. The metastable prion domain of Pin3 is the only known amyloid-based prion in *S. cerevisiae* to demonstrate very little correlation for its prion-like compositional biases, indicating some selection pressure for composition of a different sort, that nonetheless may preserve prion-forming ability (Tables 1–2).

The other three cases (Mot3p, Rnq1p and Nup100p) have either more recent ancestry as novel prion-like domains within *Saccharomycetes* (in the case of Mot3p and Rnq1p), or they arise sporadically in fungal species (Nup100p) (*An, Fitzpatrick & Harrison, 2016*; *Su & Harrison, 2019*). These three are thus not expected to demonstrate many significant correlations with measures of compositional bias, but nonetheless we see a mild negative correlation for the fLPS compositional bias of Rnq1p and Mot3p versus %Q in the proteome (Table 1), which is not typical of the other prion-forming proteins, suggesting selection pressures against Q bias in these evolutionarily recently emergent proteins.
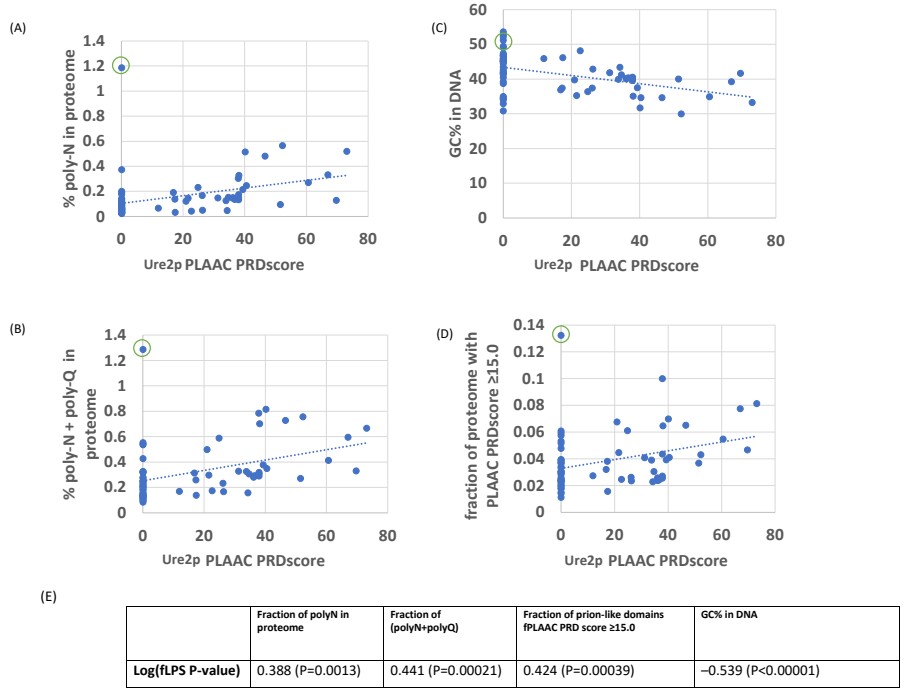

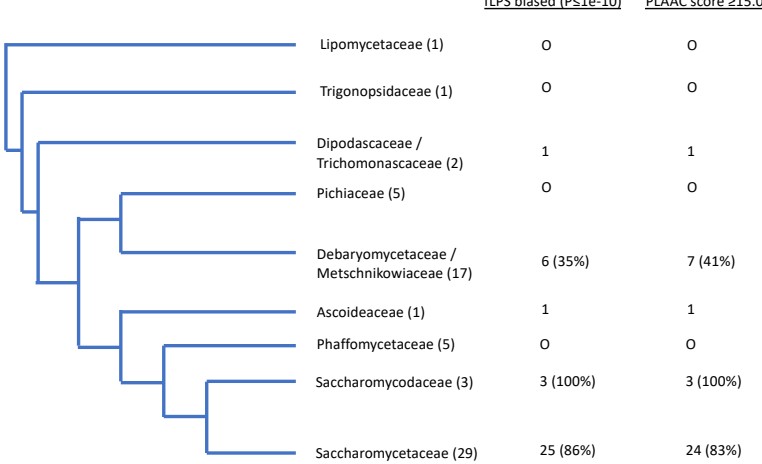

**Figure 2** **As in Fig. 1, except versus the individual PLAAC PRDscore in the Ure2p prion-forming domain.** The outlier proteome *A. rubescens* is ringed. (A) Percentage of poly-N residues in the proteome. (B) Percentage of (poly-N + poly-Q) residues in the proteome. (C) DNA GC%. (D) Fraction of proteome with PLAAC score ≥ 15.0. (E) Table of correlations and significances for plots (A) to (D).

**Figure 3** **Schematic evolutionary tree for Ure2p, showing the distribution of orthologs with prion-like composition in different evolutionary families in the Uniprot reference set of fungal proteomes.** The number of species in each family is given in brackets. The numbers of orthologs that are have fLPS P-value ≤ 1e-10 and PLAAC score ≥ 15.0 are listed in columns.

There is one species that is often a far outlier when these trends are examined, *Ascoidea rubescens* (see for example, for Ure2p in Figs. 1–2), an uncharacterized species that is the sole member of the family *Ascoideaceae*, which is geographically widely distributed and typically grows in beetle galleries in dead wood. It has a very high proportion of poly-N-rich proteins (Tables 1–2). Removal of this outlier species from the correlation analysis causes a substantial increase in correlations with %N and %poly-N, but not for %GC in DNA.

Thus, the three *S. cerevisiae* prion-forming proteomes Ure2p, Cyc8p and Swi1p appear to absorb the general mutational trends linked to the surge in formation of prion-like domains, that was observed previously (*An, Fitzpatrick & Harrison, 2016*). This trend is linked to a general decrease in %GC in the DNA (Tables 1–2).

Two other separately studied prion-forming domains are from New1p and Pub1p (*Li et al., 2014*; *Osherovich & Weissman, 2001*). These are both strongly correlated proteome-bias absorbers, with Pub1p (which is a hub for protein interaction with other prion-like proteins (*Harbi & Harrison, 2014b*) uniquely amongst all of the prion-forming domains displaying a strong correlation for both poly-N and poly-Q (Tables 1–2). Pub1p is strongly correlated despite having a low number of orthologous prion domains that have high bias for N and Q residues (53% with fLPS *P*-value ≤1e–10; Table S1) indicating that there is still correlated behavior for the weaker N/Q biases for this protein. Other prion-forming domains observed in the analysis of *Alberti et al. (2009)*, also display a similar spectrum of bias absorption across *Saccharomycetes* evolution (Table S1). Highly-correlated bias absorbers from this data whose prion-like domains are widespread in *Saccharomycetes* include Lsm4p and Gln3p, whereas other widespread prion-like domains show little or no correlation, such as Ngr1p (Tables S1, S2).

Compared to the results for N/Q-compositional bias calculated using fLPS (Table 1), the trends for prion-like composition calculated using the PRDscore from PLAAC, are similar except that New1p loses many significant correlations, and an increased correlation is captured for Sup35p *versus* the general mutational trends linked to the large-scale surge in formation of prion-like domains (*An, Fitzpatrick & Harrison, 2016*). Similar trends for PLAAC are observed if Spearman correlation coefficients are applied (by reason of some proteins having several 0.0-value PLAAC PRDscores in orthologs) (Table S3).

The above analysis uses the PLAAC PRDscore, to define the amount of prion-like composition in a bounded region, and so reflecting more absorption of biases in a way analogous to the working of the fLPS algorithm (*Harrison, 2017*; *Lancaster et al., 2014*). The PLAAC log-likelihood ratio (LLR) score has been used in the literature to pick out the most likely prion-forming sequence window within proteins (*Alberti et al., 2009*; *An, Fitzpatrick & Harrison, 2016*; *Sideri et al., 2017*; *Tetz & Tetz, 2018*). Despite the restriction of a window of fixed size (41 amino-acid residues), these LLR scores also demonstrate a similar spectrum of bias absorption, with both strong and weak absorbers evident, albeit generally with less significance (Table S4).

**Table 1** **Table for a set of known prion-forming domains of the correlations (weighted and unweighted) between the compositional bias (−log[fLPS P-value]), and a variety of parameters.** Weighted correlations are the upper value in each cell, unweighted the lower value. Where removal of the common far outlier species *Ascoidea rubescens* causes increased significance for any correlation, the third and fourth rows in a cell display the correlation coefficients (in italics). For proteins which do not have an ortholog from *Ascoidea rubescens*, the name is labelled with '††'. If its removal causes no improvement in correlations, it is labelled with '†'. Correlations significant at ≥0.0005 are labelleda and in bold, significant at <0.0005 and ≥0.0016 labelled ** and underlined, and <0.0016, and ≥0.05 are labelled *). The threshold 0.0016 comes from a Bonferroni correction to allow for the fact that 31 sequences are being tested for a correlation against any specific proteome-wide property. In column one, the name is styled according to the most significant correlation.

| Protein (Number of orthologs in brackets) | Weighted correlation (Y/N)? | A. rubescens excluded (Y/N)? | %N in proteome | %Q in proteome | %poly-N | %poly-Q | %poly-Q + %poly-N | DNA GC% | Fraction of N/Q-rich proteins in the proteome by fLPS bias threshold P-value | | |
|---|---|---|---|---|---|---|---|---|---|---|---|
| | | | | | | | | | ≥ 1e−08 | ≥ 1e−10 | ≥ 1e−12 |
| **Known amyloid-based prions in *S. cerevisiae*** | | | | | | | | | | | |
| Sup35 P05453 (62) | Y | N | 0.237 | 0.042 | 0.132 | 0.415** | 0.278* | −0.351* | 0.218 | 0.202 | 0.187 |
| | N | N | 0.136 | 0.060 | 0.035 | 0.380* | 0.180* | −0.263* | 0.142 | 0.119 | 0.095 |
| | Y | Y | *0.350* | *0.019* | *0.315* | *0.411* | *0.409** | *−0.388* | *0.348* | *0.353* | *0.366* |
| | N | Y | *0.316* | *0.021* | *0.307* | *0.370* | *0.385* | *−0.323* | *0.348* | *0.356* | *0.368* |
| **Swi1†† P09547 (56)** | Y | N | **0.661*** | −0.149 | **0.603*** | 0.074 | **0.544*** | **−0.498*** | **0.643*** | **0.627*** | **0.607*** |
| | N | N | **0.628*** | −0.184 | **0.570*** | 0.016 | **0.473*** | **−0.510*** | **0.600*** | **0.586*** | **0.568*** |
| **Cyc8 P14922 (61)** | Y | N | 0.387* | 0.292* | 0.320* | 0.361* | 0.409** | **−0.472*** | 0.398** | 0.385* | 0.364* |
| | N | N | 0.251 | 0.320* | 0.165 | 0.305* | 0.254* | −0.305* | 0.240 | 0.234 | 0.225 |
| | Y | Y | *0.522*** | *0.278* | *0.577*** | *0.354* | *0.567*** | *−0.507*** | *0.563*** | *0.581*** | *0.595*** |
| | N | Y | *0.368* | *0.307* | *0.350* | *0.297* | *0.382* | *−0.334* | *0.374* | *0.394* | *0.418** |
| **Ure2 P23202 (66)** | Y | N | **0.571*** | 0.241 | **0.468*** | 0.357* | **0.527*** | **−0.570*** | **0.556*** | **0.535*** | **0.495*** |
| | N | N | **0.485*** | 0.253* | **0.420*** | 0.330* | **0.476*** | **−0.478*** | **0.470*** | **0.453*** | **0.423*** |
| | Y | Y | *0.682*** | *0.246* | *0.676*** | *0.361* | *0.651*** | *−0.584*** | *0.687*** | *0.696*** | *0.690*** |
| | N | Y | *0.566*** | *0.259* | *0.590*** | *0.332* | *0.563*** | *−0.484*** | *0.568*** | *0.576*** | *0.572*** |
| Rnq1† P25367 (26) | Y | N | 0.139 | −0.381 | 0.096 | −0.193 | 0.037 | −0.080 | 0.081 | 0.053 | 0.010 |
| | N | N | 0.230 | −0.431* | 0.159 | −0.197 | 0.090 | −0.159 | 0.070 | 0.040 | 0.001 |
| Mot3† P54785 (25) | Y | N | 0.460* | −0.420* | 0.395 | 0.371 | 0.439* | −0.468* | 0.385 | 0.386 | 0.399* |
| | N | N | 0.393 | −0.507* | 0.264 | 0.268 | 0.299 | −0.409* | 0.140 | 0.129 | 0.146 |
| Nu100† Q02629 (11) | Y | N | 0.154 | 0.107 | −0.110 | −0.105 | −0.518 | −0.008 | −0.012 | −0.027 | −0.047 |
| | N | N | 0.224 | 0.148 | −0.058 | 0.013 | −0.499 | −0.090 | −0.017 | −0.030 | −0.042 |
| Pin3† Q06449 (55) | Y | N | 0.198 | 0.022 | 0.230 | 0.000 | −0.121 | −0.183 | 0.209 | 0.198 | 0.179 |
| | N | N | 0.179 | −0.030 | 0.200 | −0.014 | −0.046 | −0.169 | 0.165 | 0.153 | 0.137 |

*(continued on next page)*

Harrison (2020), *PeerJ*, DOI 10.7717/peerj.9669

**Table 1** (*continued*)

| Protein (Number of orthologs in brackets) | Weighted correlation (Y/N)? | *A. rubescens* excluded (Y/N)? | %N in proteome | %Q in proteome | %poly-N | %poly-Q | %poly-Q + %poly-N | DNA GC% | Fraction of N/Q-rich proteins in the proteome by fLPS bias threshold *P*-value | | |
|---|---|---|---|---|---|---|---|---|---|---|---|
| **Other prion-forming domains discussed in the text** | | | | | | | | | | | |
| **New1†† Q08972 (63)** | Y | N | **0.566\*\*\*** | 0.269* | **0.476\*\*\*** | 0.191 | **0.482\*\*\*** | **−0.482\*\*\*** | **0.513\*\*\*** | **0.501\*\*\*** | **0.486\*\*\*** |
| | N | N | **0.521\*\*\*** | 0.261* | **0.442\*\*\*** | 0.188 | **0.439\*\*\*** | **−0.449\*\*\*** | **0.468\*\*\*** | **0.458\*\*\*** | **0.446\*\*\*** |
| **Pub1 P32588 (62)** | Y | N | **0.469\*\*\*** | 0.365* | **0.484\*\*\*** | **0.707\*\*\*** | **0.686\*\*\*** | **−0.547\*\*\*** | **0.545\*\*\*** | **0.545\*\*\*** | **0.533\*\*\*** |
| | N | N | **0.457\*\*\*** | 0.243 | 0.426\*\* | **0.620\*\*\*** | **0.597\*\*\*** | **−0.532\*\*\*** | **0.449\*\*\*** | **0.448\*\*\*** | **0.442\*\*\*** |
| | Y | Y | *0.466\*\*\** | *0.401\*\** | *0.551\*\*\** | *0.734\*\*\** | *0.728\*\*\** | *−0.534\*\*\** | *0.567\*\*\** | *0.584\*\*\** | *0.594\*\*\** |
| | N | Y | *0.450\*\*\** | *0.278\** | *0.459\*\*\** | *0.646\*\*\** | *0.622\*\*\** | *−0.518\*\*\** | *0.447\*\*\** | *0.459\*\*\** | *0.471\*\*\** |

Harrison (2020), *PeerJ*, DOI 10.7717/peerj.9669

**Table 2** **Table for a set of known prion-forming domains of the correlations (both weighted and un-weighted) between the prion-like composition (PLAAC PRD-score) and a variety of parameters.** Weighted correlations are the upper value in each cell, unweighted the lower value. Where removal of the common far outlier species *Ascoidea rubescens* causes increased significance for any correlation, the third and fourth rows in a cell display the correlation coefficients (in italics). For proteins which do not have an ortholog from *Ascoidea rubescens*, the name is labelled with '††'. If its removal causes no improvement in correlations, it is labelled with '†'. Correlations significant at ≥ 0.0005 are labelled *** and in bold, significant at > 0.0005 and ≥ 0.0016 labelled ** and underlined, and > 0.0016, and ≥ 0.05 are labelled *). The threshold 0.0016 comes from a Bonferroni correction to allow for the fact that 31 sequences are being tested for a correlation against any specific proteome-wide property. In column one, the name is styled according to the most significant correlation.

| Protein (Number of orthologs in brackets) | Weighted correlation (Y/N)? | A. rubescens excluded (Y/N)? | %N in proteome | %Q in proteome | %poly-N | %poly-Q | %poly-Q + %poly-N | DNA GC% | >0.0 | Fraction of prion-like proteins in the proteome by PLAAC PRDscore ≥15.0 | ≥30.0 |
|---|---|---|---|---|---|---|---|---|---|---|---|
| **Known amyloid-based prions in *S. cerevisiae*** | | | | | | | | | | | |
| **Sup35 P05453 (62)** | Y | N | 0.292* | 0.268* | 0.174 | 0.423 ** | 0.313* | −0.345* | **0.429*** | 0.369* | 0.273* |
| | N | N | 0.160 | 0.254* | 0.040 | 0.372* | 0.181 | −0.252* | 0.307* | 0.224 | 0.108 |
| | Y | Y | *0.457*** | *0.245* | *0.437*** | *0.363*ature* | *0.497*** | *−0.401 **** | *0.574*** | *0.560*** | *0.525*** |
| | N | Y | *0.407 **** | *0.215* | *0.411 **** | *0.363*** | *0.454*** | *−0.336*** | *0.528*** | *0.506*** | *0.461*** |
| **Swi1†† P09547 (56)** | Y | N | **0.475*** | −0.206 | **0.451*** | 0.074 | 0.414** | **−0.471*** | **0.460*** | **0.464*** | 0.442** |
| | N | N | **0.465*** | −0.200 | 0.431** | 0.054 | 0.375* | **−0.470*** | 0.443** | 0.441** | 0.411* |
| **Cyc8 P14922 (61)** | Y | N | 0.353* | 0.250 | 0.325* | 0.421 ** | **0.438*** | **−0.458*** | **0.563*** | **0.486*** | 0.375* |
| | N | N | 0.244 | 0.285* | 0.183 | 0.356* | 0.288* | −0.301* | **0.462*** | 0.385* | 0.274* |
| | Y | Y | *0.453*** | *0.242* | *0.535*** | *0.417**** | *0.569*** | *−0.482*** | *0.645*** | *0.608*** | *0.548*** |
| | N | Y | *0.328*** | *0.279*** | *0.324*** | *0.353*** | *0.389*** | *−0.319*** | *0.544*** | *0.503*** | *0.429**** |
| **Ure2 P23202 (66)** | Y | N | **0.495*** | 0.151 | 0.388** | 0.308* | **0.441*** | **−0.539*** | **0.494*** | **0.424*** | 0.314* |
| | N | N | **0.448*** | 0.087 | 0.369* | 0.246* | 0.401** | **−0.453*** | 0.388** | 0.333* | 0.226 |
| | Y | Y | *0.683*** | *0.130* | *0.704*** | *0.297*** | *0.645*** | *−0.586*** | *0.627*** | *0.615*** | *0.567*** |
| | N | Y | *0.594*** | *0.071* | *0.631*** | *0.239* | *0.548*** | *−0.483*** | *0.484*** | *0.465*** | *0.393 **** |
| Rnq1† P25367 (26) | Y | N | −0.001 | −0.264 | −0.046 | −0.267 | −0.107 | 0.035 | −0.128 | −0.128 | −0.133 |
| | N | N | 0.079 | −0.340 | 0.005 | −0.242 | −0.058 | −0.027 | −0.137 | −0.139 | −0.199 |
| Mot3† P54785 (25) | Y | N | 0.149 | −0.166 | 0.135 | 0.314 | 0.196 | −0.159 | 0.114 | 0.212 | 0.283 |
| | N | N | 0.153 | −0.336 | 0.057 | 0.213 | 0.103 | −0.172 | −0.107 | −0.014 | −0.046 |
| Nu100† Q02629 (11) | Y | N | 0.090 | 0.283 | −0.185 | −0.030 | −0.165 | −0.003 | 0.304 | 0.142 | 0.021 |
| | N | N | 0.169 | 0.303 | −0.122 | 0.075 | −0.083 | −0.084 | 0.290 | 0.160 | −0.024 |
| Pin3† Q06449 (55) | Y | N | 0.000 | 0.282* | 0.025 | 0.222 | 0.121 | −0.081 | 0.112 | 0.113 | 0.159 |
| | N | N | −0.010 | 0.226 | 0.005 | 0.202 | 0.100 | −0.067 | 0.006 | 0.029 | 0.056 |

Harrison (2020), *PeerJ*, DOI 10.7717/peerj.9669

**Table 2** (*continued*)

| Protein (Number of orthologs in brackets) | Weighted correlation (Y/N)? | *A. rubescens* excluded (Y/N)? | %N in proteome | %Q in proteome | %poly-N | %poly-Q | %poly-Q + %poly-N | DNA GC% | >0.0 | Fraction of prion-like proteins in the proteome by PLAAC PRDscore ≥15.0 | ≥30.0 |
|---|---|---|---|---|---|---|---|---|---|---|---|
| **Other prion-forming domains discussed in the text** | | | | | | | | | | | |
| New1†† Q08972 (63) | Y | N | 0.368* | 0.236 | 0.301* | 0.183 | 0.326* | −0.369* | 0.412* | 0.377* | 0.299* |
| | N | N | 0.339* | 0.250* | 0.288* | 0.226 | 0.327* | −0.368* | 0.419* | 0.380* | 0.291* |
| **Pub1 P32588 (62)** | Y | N | 0.226 | **0.521***  | 0.300* | **0.756***  | **0.559***  | −0.279* | **0.597***  | **0.605***  | **0.570***  |
| | N | N | 0.241 | 0.424 ** | 0.255* | **0.679***  | **0.479***  | −0.309* | **0.504***  | **0.509***  | **0.465***  |
| | Y | Y | *0.232* | ***0.540*** | *0.381*  | ***0.771*** | ***0.628*** | *−0.274*  | ***0.631*** | ***0.674*** | ***0.695*** |
| | N | Y | *0.247* | ***0.443*** | *0.290*  | ***0.703*** | ***0.535*** | *−0.303*  | ***0.532*** | ***0.570*** | ***0.571*** |

**Prion-like N/Q-rich regions generally maintain lower lysine content than the rest of the proteome in *Saccharomycetes***

It was checked whether the N/Q-rich regions are also rich in lysine, which is encoded by AT%-rich codons, like N (asparagine). Lysine has low prion formation propensity and charged residues are disruptive to prion formation and have low prion formation propensity (*Lancaster et al., 2014*; *Osherovich & Weissman, 2001*). Lysine is a disorder-promoting residue (*Oldfield & Dunker, 2014*) and some intrinsically disordered regions have high positive charge (*Hatos et al., 2020*; *Necci et al., 2018*). However, the N/Q-rich regions consistently in general have lower lysine content that the remainder of the *Saccharomycetes* proteomes (Fig. 4). That is, the vast majority of *Saccharomycetes* species (∼98%) are below the $x=y$ line on the scatter plot (Fig. 4A). This is also obvious in the distributions of K fraction (Fig. 4B, values for prion-formers are lower, $t$-test P $=$ 1e−140). Thus, these regions are not simply absorbing higher levels of AT% in their DNA through the embedding within them of amino-acid residues encoded by codons with high AT%.

## DISCUSSION

These results indicate that compositional aspects of many individual prion-formers behaved in a correlated way in relation to general trends as they panned out over millions of years across various sub-clades. Also, this surge in prion-like region formation is directly linked to a general trend for GC% decrease across the *Saccharomycetes* clade. However, some prion-forming domains resist the absorption of such mutational trends, such as the meta-stable prion-former Lsb2/Pin3 (*Chernova et al., 2017b*), despite it being as widely conserved as a protein as those that more easily absorb biases, such as Cyc8p and Swi1p. This suggests some greater selection pressure on amino-acid composition. The Sup35p prion-forming domain also shows some special behavior: demonstrating a stronger correlation between overall proteome poly-Q levels and its own N or Q compositional bias as determined by the program fLPS. The Sup35 prion-forming domain has a subdomain with specific local patterns involving Q residues that is required for chaperone-dependent prion maintenance, that is separate from the N-terminal N/Q-rich region that is necessary for prion nucleation and fibre growth (*MacLea et al., 2015*). Also, the Sup35 prion-like domain has a more ancient origin before the last common ancestor of *Saccharomycetes*, and outside this clade it tends to have a predominant Q-bias that has been maintained within *Saccharomycetes*, resisting the trend for greater N-bias (*An, Fitzpatrick & Harrison, 2016*). However, this is also the behaviour of Cyc8p and Swi1p outside of *Saccharomycetes* (*An, Fitzpatrick & Harrison, 2016*), so this result is demonstrating an evolutionary behavior peculiar to Sup35p.

The Pub1p prion-forming domain shows strong correlations for both Q and N bias indicators. It is possible that proteins such as Pub1p that interact a lot with other prion-like proteins (*Harbi & Harrison, 2014b*) 'need' to absorb more general compositional trends so that they can promiscuously bind with a large list of partners. Prion-like aggregation has been shown for both Pub1p in yeast and for its co-ortholog Tia1 in humans (*Gilks et al., 2004*; *Li et al., 2014*). Its prion-like composition has also largely been maintained since the

(A)

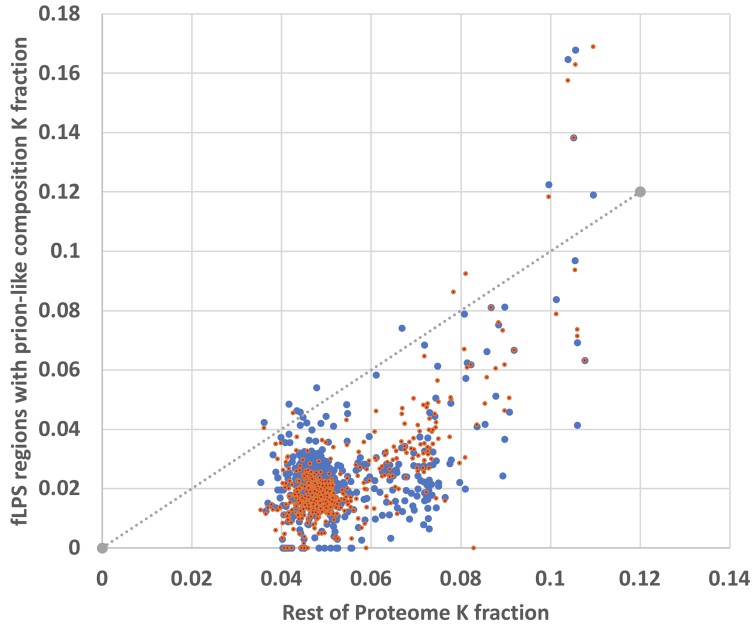

(B)

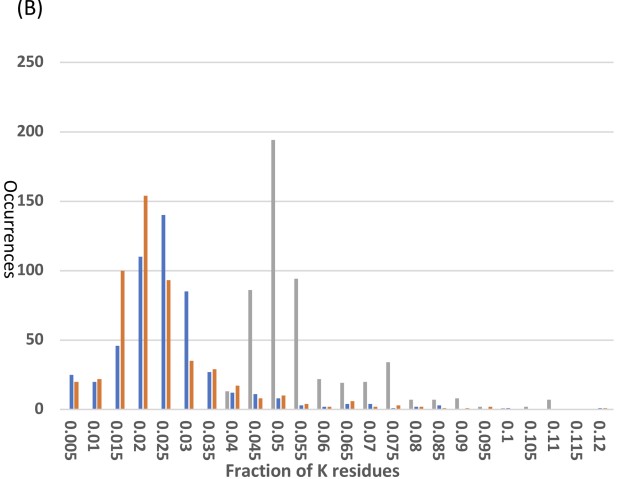

**Figure 4 Analysis of fraction of K (lysine) residues.** (A) Scatter plots of fraction of K within the N/Q-rich regions of prion-forming domains plotted versus the overall fraction of K in proteomes. Blue points are for the set of known amyloid-based prions in Fig. 4, and orange points for the total list of prion-forming domains including those listed in Table S2. The *x=y* line is indicated. (B) Histograms for the data in (A). The same colour scheme is kept, except that the data for the rest of the proteome is in grey. Each bin is labelled with its higher bound.

last common ancestor of eukaryotes (*Su & Harrison, 2020*). Thus, its strong absorption of mutational trends for Q and N residues has not been a barrier to such a conservation of prion-like composition.

The methodology applied here might also be useful in the analysis of human proteins with N/Q biases, such as those linked to amytrophic lateral sclerosis or huntingtin from

Huntington's disease (*An & Harrison, 2016*; *Monahan et al., 2018*), or to other non-N/Q-biased prion-forming domains, such as in alpha-synuclein (*Watts, 2019*). In particular, prion-forming domains from any such proteins that display little or no correlation with general compositional trends in the proteome may be under selection pressure against aggregation, or for a functional role for which compositional sequence parameters are precisely modulated. Recent research suggests that sequence mutations leading to subtle amino-acid side-chain differences in a short disordered segment of the Sup35p prion-forming domain alter its conformational preferences and markedly modify its cross-reactivity with infectious prion seeds (*Shida et al., 2020*). Such subtle effects are interesting in light of the fact that prion formation is largely governed by compositional preferences (*Cascarina et al., 2018*; *Ross, Baxa & Wickner, 2004*; *Toombs, McCarty & Ross, 2010*). Given such considerations, our results imply that some specific segments of prion-forming domains may be under selective constraint, while other segments are more free to absorb large-scale mutational trends, such as the surge in asparagine-rich prion-like tracts during *Saccharomycetes* speciation (*An & Harrison, 2016*).

One form of selective constraint was examined in detail, the avoidance of lysine residues. Both asparagine and lysine are encoded by an AT-rich codon repertoire that just differs at the third codon position. Naively one would expect them to co-occur, since lysine has disorder-promoting character and prion-forming domains are intrinsically disordered (*Harbi & Harrison, 2014a*; *Harbi et al., 2012*). However, lysine has low prion formation propensity and charged residues are disruptive to prion formation and have low prion formation propensity (*Lancaster et al., 2014*; *Osherovich & Weissman, 2001*). Here, we observed that lysine residues are avoided as *S. cerevisiae* prion-forming domains evolve across *Saccharomycetes*. Further development of such co-occurrence analysis for amino-acid residue types might yield further clues about the conservation of prion-forming status or other selective constraints on amino-acid composition in protein regions of unknown character (*Harrison, 2018*).

The results here provide a case study of mutational trend absorption by disordered regions generally. The results suggest some methodology for analyzing selection pressures on individual intrinsically disordered regions within the context of the behaviour of other sequences from the same proteome.

## CONCLUSIONS

Many prion-forming domains, and intrinsically disordered regions generally, are continually absorbing overall mutational trends in their proteomes, but this is modulated by specific selection pressures. A spectrum of bias absorption is observed from Lsb2/Pin3—which appears refractive to the mutational trends and shows little or no correlation—to Pub1, which shows very strong correlation to both asparagine- and glutamine-based biases.

The present analysis can be seen as a case study of the absorption of mutational trends in compositionally biased domains. The *S. cerevisiae* prion-forming list of proteins is particularly well-suited for this. Firstly, there is a substantial set of them that has accumulated via experimental analysis over the past two decades. Secondly, within the

*Saccharomycetes* there has been a wholesale shift in mutational trends relative to other fungi over the past hundreds or millions of years, which provides the major context for their molecular evolution. This work can be expanded to analyze further these phenomena on a larger scale.

### Funding

This work was supported by the Natural Sciences and Engineering Research Council of Canada. The funders had no role in study design, data collection and analysis, decision to publish, or preparation of the manuscript.

### Grant Disclosures

The following grant information was disclosed by the author:
Natural Sciences and Engineering Research Council of Canada.

### Competing Interests

The author declares there are no competing interests.

### Author Contributions

- Paul M. Harrison conceived and designed the experiments, performed the experiments, analyzed the data, prepared figures and/or tables, authored or reviewed drafts of the paper, and approved the final draft.

### Data Availability

The amino-acid sequences of the orthologs sets of proteins with prion-forming domains in *S. cerevisiae* are available in Data S1.

### Supplemental Information

Supplemental information for this article can be found online at http://dx.doi.org/10.7717/peerj.9669#supplemental-information.

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
