# Peer review of "Variable absorption of mutational trends by prion-forming domains during Saccharomycetes evolution"

_PeerJ, doi:10.7717/peerj.9669_

## Round 0.1 · original submission · Major Revisions

· Academic Editor

Major Revisions

Dear Dr. Harrison:

Thanks for submitting your manuscript to PeerJ. I have now received two independent reviews of your work, and as you will see, the reviewers raised some concerns about the research. Despite this, these reviewers are optimistic about your work and the potential impact it will have on research studying prions and their role in yeast biology. Thus, I encourage you to revise your manuscript, accordingly, taking into account all of the concerns raised by both reviewers.

While the concerns of the reviewers are relatively minor, this is a major revision to ensure that the original reviewers have a chance to evaluate your responses to their concerns.

There are many comments by both reviewers that ask for more information on specific issues; please address these. Please consider changing certain file types and also including some references deemed missing by the reviewers. Please also consider expanding the Discussion a bit, per reviewer 2’s suggestions.

I look forward to seeing your revision, and thanks again for submitting your work to PeerJ.

Good luck with your revision,

-joe

Reviewer 1 ·

Basic reporting

The investigation touches on the very interesting topic of the evolution of amyloid and prion-forming domains. Nevertheless, number issues can be improved before publication.

1. Abbreviations for yeast prion names (PSI, PIN, SWI, etc) should be written in italic ([/PSI/+], [/PIN/+], [/SWI/+]). The yeast prion forming nucleoporin was originally called Nup100 but not Nu100 (Halfmann R et al. 2012. Prion formation by a yeast GLFG nucleoporin. Prion 6: 391–399.).

2. I would recommend simplifying the phrase in the introduction, lines 86-90.
"It is discovered that these protein regions have a spectrum of behavior, variably absorbing mutation biases that are observable in the proteome as a whole, evidenced in the numbers of prion-like proteins, the % guanidine and cytidine (GC%) in the DNA, and the proportions of poly-asparagine and poly-glutamine."

3. Tables 4-5 referenced on line 169 are absent.

4. Couple mistakes or missed references were found.
line 38. The articles demonstrating the role of Sup35 in translation termination should be mentioned:
Stansfield et al. 1995. The products of the SUP45 (eRF1) and SUP35 genes interact to mediate translation termination in Saccharomyces cerevisiae. EMBO J 14: 4365–4373.
Zhouravleva et al. 1995. Termination of translation in eukaryotes is governed by two interacting polypeptide chain release factors, eRF1 and eRF3. EMBO J 14: 4065–4072.
line 53. The reference Alberti et al., 2009 might be added to the list of algorithms (Alberti et al. 2009. A systematic survey identifies prions and illuminates sequence features of prionogenic proteins. Cell 137: 146–58).

5. The tables in the supplementary are hard to follow because they are presented as images. Also, I'd suggest presenting protein names in a separate column, which will contain the only name of the protein but not merged UniprotID, protein name, and organism.

6. I'd suggest adding two new columns ("Weighted correlation" and "Ascoidea rubescens excluded") with "Yes/No" values in tables 1 and 2 because it is a little bit difficult to catch the difference between 2-4 values in one cell in tables 1-2.

7. The main idea of Figure 4, as I understand, is a demonstration that lysine frequency within prion forming domains is lower than in the proteome. From this point of view, the boxplot or barplot is more informative. Also, the demonstration that this fact is specific for NQ-rich domains will increase the impact of the paper. If the lysine frequency in non-NQ-rich region is comparable to the same parameter across proteome, corresponding speculation can be made.

8. The author attached the sequences of S. cerevisiae proteins, which were analyzed. The addition of ortholog sequences can increase the interest to the paper.

Experimental design

The choice of correlation method (Pearson correlation) for calculations should be explained and it is essential to check that the data are normally distributed. Otherwise, I would advise using the Spearman correlation coefficient.

Validity of the findings

The author demonstrated that NQ-rich proteins behave as a whole proteome in terms of accumulation of asparagine and glutamine and claimed that these proteins adsorb the general trend. This observation seems to be expected, however, it has not been published yet. At the same time revealed correlations raise a new question about the specificity of this feature for NQ-rich proteins/domains only. If the same results can be obtained for a randomly chosen yeast protein the conclusion about the "absorption of general mutational trend" may be preliminary.

·

Basic reporting

This is a concise and well-written manuscript which will be of interest to, and a resource for, researchers working on prion proteins, yeast biology, intrinsically disordered proteins and computational biology more broadly. The study builds on work on prion phenomena from the author's laboratory over the past decade. The overall conclusion of the paper is that prion-forming domains in a number of yeast proteins (from species in the Saccharomycetes class) "have a spectrum of behaviour, variably absorbing mutation biases that are observable in the proteome as a whole".
*Suggestion #1: It might help, at the outset of the introduction, to have a brief sentence to highlight that Saccharomycetes is a class of fungi that includes species such as S. cerevisiae, given that yeast prions are usually (at least initially) associated first with proteins in this species.
*Suggestion #2: Would it be possible to have the subsection headings in the Results section be more descriptive, to include the essence of the findings in each subsection?

Experimental design

The methodology and analysis are clear. Data file S1 provides the prion-forming domains used in the manuscript in FASTA format.
*Suggestion #3: If possible, it might be useful to convert the formats of Tables S1/S2/S3 into text-based or Word document tables, so that the text, UniProt IDs, etc., could be easily searched.
*Suggestion #4: In Figures 1 and 2, it might help to include "Ure2p" in the x-axis labels, and, in the figure legends, to mention (and perhaps also mark on the figures) the outlier in panels A, B and D.
*Suggestion #5: Eight measures of compositional bias are stated starting from line 110. It might help to explain in a few additional sentences, why, in the context of (yeast) prion research and protein structure disorder, these measures were selected and if other measures could also be used and why they might not have been relevant to this study.

Validity of the findings

The underlying data has been provided and the conclusions and methodology link back to the original question and scope of the paper. The findings have also been presented with reference to relevant prior publications. For example, citing the reference An et al. 2016, the author states in the Introduction that "A large population of yeast-prion-like proteins emerged en masse early in Saccharomycetes evolution, as a result of mutational trends to form more polyasparagine runs, thus providing an evolutionary 'test set' from which several prion-forming domains seem to have developed". In the Results section, when discussing this theme, the author writes that "during the surge in formation of prion-like regions during Saccharomycetes evolution, the degree of N-bias in the individual prion-former Ure2p also increased in correlation with the general trend as it panned out across various sub-clades". These linkages are quite helpful. However, I think there could be some further commentary and speculation added to the Discussion section in terms of the broader context of the findings and what steps could be taken in subsequent studies (please see next section of comments).

Additional comments

In addition to the suggestions above, I would like to suggest the following three points to be considered for possible addition to the Discussion section:
*Suggestion #6: Given that there is now considerable interest in the prion-like behavior of alpha-synuclein, which is thought to be a disordered protein (perhaps notwithstanding its plasma membrane interactive domains), there might be some interest and relevance in mentioning this in the Discussion.
*Suggestion #7: Somewhat similar to the above, might there be some interest in mentioning the possibility of applying aspects of the methodology of this paper to computationally investigating polyglutamine tracts in Huntington's disease and other neurodegenerative conditions?
*Suggestion #8: Given that the aim of the manuscript is centered on prion-forming domains and disordered regions, it might be helpful to provide some speculation as to whether one could use the study of selection pressures on the sequences of these regions to understand their *structure* a bit better. Just as an example, let us briefly consider this scenario: Protein X is a prion-former and, (i) is intrinsically disordered, and (ii) is capable of forming amyloids. If we say that protein X, because of its intrinsic disorder, is "unfolded" in the cell, does this mean that *each* protein X in the cell is unfolded in a random and *unique* way and therefore different from another protein X in its structure? Or, do we mean that all protein X's in the cell are "unfolded" in the same way? I think the latter scenario is more likely, because: (i) If each protein X was different structurally from another protein X in its "random structure", then how can they undergo a homologous amyloid-forming process, a process which implies aggregation based on some new and shared acquired templating structure (this might also be true in the case of selective heterologous amyloid deposits, e.g., in the case of tau protein and Abeta)? (ii) The amino acid sequence of all protein X's in the cell are the same; and (iii) the cellular microenvironment that these proteins are exposed to is also the same. A relevant recent paper to this topic is: Shida et al., "Short disordered protein segment regulates cross-species transmission of a yeast prion", Nat Chem Biol 2020.

---

## Round 0.2 · Minor Revisions

· Academic Editor

Minor Revisions

Dear Dr. Harrison:

Thanks for revising your manuscript. The reviewers are very satisfied with your revision (as am I). Great! However, there are a few minor edits to make. Please address these ASAP so we may move towards acceptance of your work.

Best,

-joe

Reviewer 1 ·

Basic reporting

All my comments were addressed, thank you. Two last advises.
1. Please, add the full legend on fig.4B, including all three colors, or hide the legend on the plot.
2. The square brackets in the prion names is usually writtern in general font and "+" is in superscript.

Experimental design

All my comments were addressed, thank you.

Validity of the findings

All my comments were addressed, thank you.

·

Basic reporting

The suggestions in the previous review have been implemented in the revised version.

Experimental design

The suggestions in the previous review have been implemented in the revised version.

Validity of the findings

The suggestions in the previous review have been implemented in the revised version.

---

## Round 0.3 · accepted · Accept

· Academic Editor

Accept

Dear Dr. Harrison:

Thanks for revising your manuscript based on the concerns raised by the reviewers. I now believe that your manuscript is suitable for publication. Congratulations! I look forward to seeing this work in print, and I anticipate it being an important resource for groups studying prions and their role in yeast biology. Thanks again for choosing PeerJ to publish such important work.

Best,

-joe